# Structural Determinants of the Specific Activities of an L-Amino Acid Oxidase from *Pseudoalteromonas luteoviolacea* CPMOR-1 with Broad Substrate Specificity

**DOI:** 10.3390/molecules27154726

**Published:** 2022-07-24

**Authors:** Kyle J. Mamounis, Maria Luiza Caldas Nogueira, Daniela Priscila Marchi Salvador, Andres Andreo-Vidal, Antonio Sanchez-Amat, Victor L. Davidson

**Affiliations:** 1Burnett School of Biomedical Sciences, College of Medicine, University of Central Florida, Orlando, FL 32827, USA; kyle.mamounis@gmail.com (K.J.M.); nogueira.mluiza@gmail.com (M.L.C.N.); 2Department of Molecular Biology, Center of Exact and Natural Sciences, Federal University of Paraiba, João Pessoa 58051-900, PB, Brazil; danimarchi@dbm.ufpb.br; 3Department of Genetics and Microbiology, University of Murcia, 30100 Murcia, Spain; andresandreoiv@gmail.com (A.A.-V.); antonio@um.es (A.S.-A.)

**Keywords:** enzyme, flavoprotein, hydrogen peroxide, kinetics, phylogenetic relationship, protein structure–function

## Abstract

The *Pseudoalteromonas luteoviolacea* strain CPMOR-1 expresses a flavin adenine dinucleotide (FAD)-dependent L-amino acid oxidase (LAAO) with broad substrate specificity. Steady-state kinetic analysis of its reactivity towards the 20 proteinogenic amino acids showed some activity to all except proline. The relative specific activity for amino acid substrates was not correlated only with *K*_m_ or *k*_cat_ values, since the two parameters often varied independently of each other. Variation in *K*_m_ was attributed to the differential binding affinity. Variation in *k*_cat_ was attributed to differential positioning of the bound substrate relative to FAD that decreased the reaction rate. A structural model of this LAAO was compared with structures of other FAD-dependent LAAOs that have different substrate specificities: an LAAO from snake venom that prefers aromatic amino acid substrates and a fungal LAAO that is specific for lysine. While the amino acid sequences of these LAAOs are not very similar, their overall structures are comparable. The differential activity towards specific amino acids was correlated with specific residues in the active sites of these LAAOs. Residues in the active site that interact with the amino and carboxyl groups attached to the α-carbon of the substrate amino acid are conserved in all of the LAAOs. Residues that interact with the side chains of the amino acid substrates show variation. This provides insight into the structural determinants of the LAAOs that dictate their different substrate preferences. These results are of interest for harnessing these enzymes for possible applications in biotechnology, such as deracemization.

## 1. Introduction

An L-amino acid oxidase (LAAO) catalyzes the stereospecific oxidative deamination of an L-amino acid. The substrates for an LAAO are the L-amino acid, H_2_O, and O_2_, which are converted to the corresponding α-keto acid, ammonia, and H_2_O_2_. LAAOs are typically flavoenzymes that utilize a flavin adenine dinucleotide (FAD) cofactor to catalyze the reaction [1]. Alternately, there is a phylogenetically and structurally unrelated family of amino acid oxidases that produces the same products but utilizes endogenous quinone cofactors formed through post-translational modification [2] instead of a flavin.

LAAO activity was first observed in 1910 with the oxidation of L-tyrosine in perfused liver [3], and further LAAO activity was demonstrated in the rat kidney and liver in 1944 [4]. Since then, a variety of other sources and roles for LAAOs have been identified. The physiological roles of LAAOs vary across organisms, and in many cases remain to be clearly determined. Snake venom is a prolific source of studied LAAOs [5,6]. The LAAOs in snake venom induce apoptosis via the H_2_O_2_ reaction product [7]. Invertebrates dwelling in wet environments utilize LAAO-produced H_2_O_2_ as an antimicrobial agent. The giant snail *Achatina fulica* expresses LAAO in its mucus to inhibit bacterial growth [8], as does the sea hare [9,10]. Expression of LAAOs in *Pseudoalteromonas* species is promoted by genes involved in environmental monitoring and is associated with deterring the settlement of competing marine fouling organisms [11]. In *Marinomonas mediterranea* and some other bacteria, the H_2_O_2_ released is involved in biofilm differentiation [12]. *Neurospora crassa* uses LAAO to generate ammonia from L-amino acids as an alternative nitrogen source when other sources are limited [13]. A mammalian LAAO, interleukin-4-induced gene 1, has been studied in milk as a mastitis-preventing factor that is induced by the T-helper-cell-secreted cytokine interleukin-4 [14].

Most flavin-dependent LAAOs are stereospecific for L-amino acids, with some exceptions such as the one from the soil bacterium *Bacillus carotarum* that oxidizes seven D-enantiomers of its ten L-amino acid substrates [15]. Distinct flavin-dependent D-amino acid oxidases (DAAOs) oxidize the D-amino acids, and they are also fairly enantiomer specific [16,17]. In general, LAAOs are specific for certain classes of amino acids. For example, snake venom LAAOs prefer hydrophobic amino acids [6] whereas the LAAO from sea hare (*Aplysia californica*) prefers the basic amino acids L-arginine and L-lysine [9]. LAAOs from bacterial sources exhibit a broader range of substrate specificities [18], such as the LAAO that was characterized from *Rhodococcus opacus* [19,20]. Two endogenous quinone-bearing LAAOs that utilize a cysteine tryptophylquinone cofactor (CTQ) [21] have been characterized that do not oxidize a broad range of amino acids but are specific for just one: an L-lysine ε-oxidase from *M. mediterranea* [22] and glycine oxidases from *M. mediterranea* [23] and *Pseudoalteromonas luteoviolacea* [24].

In addition to producing the CTQ-dependent glycine oxidase, extracts of two strains of *P. luteoviolacea*, CPMOR-1 and CPMOR-2, were shown to possess additional broad-spectrum LAAO activity [25]. This activity was attributed to a 74-kDa flavoprotein [26]. Recently, the structure of an LAAO from *P. luteoviolacea* was reported [27]. The strain from which that protein was isolated was not indicated; however, the reported protein sequence was that of the one from strain CPMOR-2. Here, we report the expression and characterization the LAAO from strain CPMOR-1, which is distinct from the other while sharing an 86.8% identity [26]. In fact, the phylogenetic analysis reported herein indicates that these two proteins reside in different branches of the phylogenetic tree. The steady-state kinetic parameters for the reactions of the *P. luteoviolacea* CPMOR-1 LAAO with all proteinogenic and some unnatural amino acids are reported. The relative catalytic efficiency, *K*_m_, and *k*_cat_ for the amino acid substrates were analyzed in relation to the structural features of the active site of the enzyme. This allowed determination of which specific residues in the active site may dictate the relative *K*_m_ and *k*_cat_ for different amino acid substrates of this broad-range LAAO. These results and structural correlations were compared with those of LAAOs from other diverse sources with different and more narrow substrate specificities: a snake venom LAAO (SvLAAO) from *Calloselasma rhodostoma* that prefers aromatic amino acids [28] and a fungal LAAO from *Trichoderma viride* (TvLAAO) [29] that is specific for Lys. With this information, it was possible to identify structure–function relationships that define the relative substrate specificities of the different LAAOs. This information is also discussed in terms of the potential applications of *P. luteoviolacea* CPMOR-1 LAAO in biotechnology.

## 2. Results

### 2.1. Properties of P. luteoviolacea CPMOR-1 LAAO

The purified LAAO ran as a single band on SDS-PAGE at the expected molecular mass of approximately 74 kDa (data not shown). The visible absorbance spectrum of the protein exhibited overlapping peaks at 388 and 464 nm that are characteristic of the oxidized FAD cofactor (Figure 1A). The addition of amino acid substrates to the enzyme under anaerobic conditions caused bleaching of this absorbance feature to yield a spectrum characteristic of the reduced FADH_2_ (Figure 1B). Subsequent exposure to air resulted in a return to the initial spectrum consistent with reoxidation of the cofactor. Alternatively, dithionite was added to reduce the oxidized protein under anaerobic conditions (Figure 1C). This caused a similar bleaching of the visible absorbance; however, it was a biphasic process with the initial formation of an intermediate with spectral features characteristic of an FAD anionic semiquinone [30], followed by conversion to the FADH_2_ spectrum. The second phase was very slow, requiring approximately 8 min for completion. This is consistent with the reduction in FAD proceeding via two discrete one-electron transfers in which the first one-electron reduction is far more thermodynamically favorable than the second one-electron reduction. Because the stable semiquinone intermediate was not seen during substrate reduction, this indicates that the reduction by the amino acid substrate was a single-step two-electron reduction.

Comparison of the amino acid sequences of the LAAOs from *P. luteoviolacea* CPMOR-1 and CPMOR-2 indicated that they exhibit an 86.8% identity and 93.7% similarity. The crystal structure of the latter was reported [27]. Given the high sequence similarity, a homology model was constructed of the *P. luteoviolacea* CPMOR-1 LAAO using the structure of the *P. luteoviolacea* CPMOR-2 LAAO (PDB 7OG2) as a template (Figure 2). The pink color in the overall structure indicates regions of the protein that harbor differences in the sequence with *P. luteoviolacea* CPMOR-2 LAAO. The reliability of this model was excellent. The Global Model Quality Estimation of the model was 0.9, which is near the maximum of 1.0 and well above the threshold of 0.7 for reliability. The QMEAN Z-score (global model quality and residual) was −0.35, which is near the maximum of zero and well above the threshold of −4.0 for good quality. This is expected as the model is based on the structure of a highly similar protein with 95% coverage.

### 2.2. Steady-State Kinetic Analysis of Reactions of P. luteoviolacea CPMOR-1 LAAO with Amino Acids

Each of the 20 proteinogenic amino acids were tested as a substrate for this LAAO (Table 1). At least some activity was detectable with all amino acids except for Pro, Thr, Asp, and Gly, which had minimal activity with catalytic efficiencies (*k*_cat_/*K*_m_) less than 1. The following D-amino acids were tested: D-Trp, D-Ala, D-Leu, and D-Met. No activity for any of these was observed at up to millimolar concentrations near their solubility limits, indicating that the enzyme is highly stereospecific for LAAOs. Other amines, amides, and related molecules were also tested as substrates, and none presented any activity up to the concentrations of their solubility limits. These compounds were hydroxylamine, ethanolamine, methylamine, penicillamine, sulfanilamide, thiourea, and glutathione. Thus, *P. luteoviolacea* CPMOR-1 LAAO is very specific for L-amino acids. The amino acid substrates in Table 1 are listed in the order of their catalytic efficiency. It should be noted that the level of catalytic efficiency was not solely related to the relative value of just *K*_m_ or just *k*_cat_. The two parameters often varied independently of each other. It should also be noted that some other kinetic studies of LAAOs were performed at 37 °C. The current study was performed at 25 °C, as it is closer to the temperature at which the host bacterium usually grows. Unlike *E. coli*, *P. luteoviolacea* is not present as animal gut flora but is a marine bacterium that grows in an environment with much lower temperatures than the human body.

The amino acid substrates may be viewed in groups with similar catalytic efficiency, which can be related to the structures of the amino acid side chains. The most efficient substrates were Leu, Met, and Gln, each of which have *k*_cat_/*K*_m_ values of approximately 10^5^ M^−1^·s^−1^. As can be seen in Figure 2, a common feature of these amino acids is a non-polar side chain of a similar length. Leu has a three-carbon chain with a methyl group branching off the chain (Figure 3A). Met has two carbons followed by a sulfur, which is bonded to a methyl group (Figure 3B). Gln has a three-carbon chain that terminates with an amide (Figure 3C).

The importance of the precise structure of the amino acid side chain was evidenced by comparison of the kinetic parameters for other amino acids with similar but subtly different structures, and significantly different activity. The catalytic efficiency of Ile was approximately 100-fold less than Leu (see Table 1, Figure 3D). The subtle difference in the structure is that in Ile, the methyl group branches off the β-carbon, whereas in Leu, it is bonded to the γ carbon. The catalytic efficiency of Val, with the same β-carbon methyl group as Ile but a sidechain that is one methyl group shorter, was another 10-fold less efficient than Ile (Figure 3E). The presence of a carboxyl group on an amino acid side chain significantly decreased the catalytic efficiency. This can be seen by comparing Glu and Gln. For Glu, *K*_m_ increased 7-fold and *k*_cat_ decreased 8-fold relative to Gln (see Table 1, Figure 3F).

The three next best substrates for *P. luteoviolacea* CPMOR-1 LAAO were Phe, Trp, and Tyr. Each exhibited *k*_cat_/*K*_m_ values of approximately 10^4^ M^−1^·s^−1^ (Table 1). The *K*_m_ and *k*_cat_ values for each of these three amino acids were similar. The common structural feature of these amino acids is that the side chain of each has an aromatic group (Figure 4).

Comparison of the structures of the amino acids in Figure 3 and Figure 4 with their kinetic parameters in Table 1 reveals how the side-chain structures can differentially affect *K*_m_ and *k*_cat_. The decrease in the catalytic efficiencies of Phe, Trp, and Tyr relative to Leu, Met, and Gln is due primarily to a large decrease in *k*_cat_, with much less change in *K*_m_. This suggests that the binding affinities for the six amino acids are similar. An electronic or chemical influence of the side chains on *k*_cat_ is unlikely as the aromatic groups are distant from the substrate α-carbon and amino group that undergo reaction. Therefore, it is possible that the presence of an aromatic side chain influences the positioning of the reactive part of the substrate relative to the FAD in the active site, which decreases the rate of reaction. Conversely, the decreases in the catalytic efficiencies of Ile and Val, relative to Leu, are due primarily to increases in *K*_m_ with little change in *k*_cat_. This suggests that the shape of the branched chain negatively affects the binding affinity. As noted above, the presence of a carboxyl group on the substrate side chain significantly decreases the catalytic efficiency. This significantly increased *K*_m_ and decreased *k*_cat_ for Glu relative to Gln. Similarly, while Asn was a very poor substrate, reaction with Asp was barely detectable. In general, charged side chains tend to be poorer substrates, primarily because of very high *K*_m_ values: Arg is 2.1 mM, Lys is 43 mM, and His is 1 mM (Table 1). The value for His is much greater than for the three aromatic amino acids, again suggesting that its higher polarity may be detrimental to binding. The presence of a hydroxyl group on the side chain is also detrimental to the substrate reactivity. The structures of Cys with Ser are identical except for a terminal OH on the Ser side chain and a terminal SH on the Cys side chain. The *K*_m_ value for Ser is approximately 100-fold greater than for Cys, suggesting that it may occupy a hydrophobic substrate pocket, given the more hydrophobic character of the Cys side chain. Similarly, Thr showed an approximately 30-fold higher *K*_m_ as compared to Val, a difference that may be structurally related to the substitution of a methyl group in Val with a hydroxyl group in Thr.

Further insight into the influence of the amino acid side chain structure on reactivity was explored using unnatural amino acids (Table 2). The effect of addition of a hydroxyl substituent to the indole ring of Trp was tested using 5-hydroxy-tryptophan. A significant reduction in the catalytic efficiency was observed when the hydroxyl group was present. This was primarily due to an approximately 10-fold increase in *K*_m_ with only a modest change in *k*_cat_. In contrast, the addition of a nitro group to the ring of Tyr, as seen with 3-nitro-tyrosine, had little effect on *K*_m_ or *k*_cat_. Homocysteine was tested to see the effect of the addition of a methyl group to the end of the side chain of Cys. Again, this had little effect on the kinetic parameters, which may be related to the inherent hydrophobicity of the Cys side chain.

### 2.3. Correlation of LAAO Sequences and Structures with Substrate Specificity

To gain insight into the roles of active-site residues in influencing the specificity for particular amino acid substrates, the *P. luteoviolacea* CPMOR-1 LAAO sequence and structure were compared with those of two other LAAOs with stricter substrate specificity. The SvLAAO from *C. rhodostoma* prefers aromatic amino acids [28], as do SvLAAOs in general [6]. The fungal TvLAAO is a specific L-lysine oxidase [29]. A comparison and alignment of the amino acid sequences of these LAAOs are shown in Figure 5. When aligned, the sequence identities to *P. luteoviolacea* CPMOR-1 LAAO were 43.6% for SvLAAO and 57.5% for TvLAAO. Residues in the substrate-binding sites are highlighted for comparison and to illustrate the remarkable conservation of the relative position of these key residues in the sequences. The sequence of the related *P. luteoviolacea* CPMOR-2 LAAO is included for comparison.

The homology model of the structure of *P. luteoviolacea* CPMOR-1 LAAO was compared with those of SvLAAO and TvLAAO (Figure 6). This reveals that despite their relatively low sequence identity, the overall structures of these three LAAOs from these diverse species are quite similar. The crystal structure of SvLAAO includes a Phe substrate present in the active site and the crystal structure of TvLAAO includes a Lys substrate present in the active site. The active sites and FAD cofactor reside in similar positions on the proteins. Small insertions and deletions of some amino acids are observed, causing an increase or decrease in the size of the α-helices or β-sheets when the sequences and structural elements that form each structure are compared. Two main differences between *P. luteoviolacea* CPMOR-1 LAAO and the other structures are the insertion of an α-helix from Leu411 to Pro439, and another α-helix at the C-terminus starting at Phe590 (see Figure 5 and Figure 6).

The residues in the region of the active site that interact with the α-carbon of the amino acid substrate and the carboxyl and amino substituents in each of the three LAAOs are shown in Figure 7. These residues are highly conserved in identity and position in the three LAAOs. In each case, an Arg guanidino group is in position to interact with the carboxyl group. The hydroxyl of a nearby Tyr is also in a position to coordinate a carboxyl oxygen of the substrate. A conserved Lys is present in each structure. This residue is seen in other LAAOs and has been suggested to stabilize an ordered water in the structure, which likely participates in the catalytic reaction [31]. The residues near the amino group are either a Gly in *P. luteoviolacea* CPMOR-1 LAAO and SvLAAO, or an Ala in TvLAAO. In all three proteins, there is also a Trp conserved in this region. The presence of the Trp is consistent with the nearby amino group of the substrate being uncharged.

The region of the active site that interacts with the side chain of the amino acid substrate is shown in Figure 8. This area is more varied between proteins than those shown in Figure 7. This is expected given the differences in substrate specificity. The structures of the SvLAAO and TvLAAO have bound amino acid substrates that provide some insight into what residues interact directly with the amino acid side chain. It should be noted that residues in this region might change position when substrate is present relative to the position in the absence of substrate. This was noted for SvLAAO (Figure 8), where His241 and Arg340 exhibited different conformational states in the presence and absence of substrate [28]. The structure of *P. luteoviolacea* CPMOR-1 LAAO does not have a substrate present. As such, it may also exhibit rearrangement of some of these residues to accommodate the amino acid substrate. Since this LAAO has broad substrate specificity, it is likely that these rearrangements will be different for different substrates. The residues in this region are a combination of amino acids that are aromatic (Phe229, Phe249), hydrophobic (Leu245, Leu361, Val533), polar (Gln359, Gln535), and charged (Asp447). This provides residues that could potentially interact with a variety of amino acid substrates when properly positioned. There are some interesting differences between the SvLAAO and *P. luteoviolacea* CPMOR-1 LAAO sites. Residues in *P. luteoviolacea* CPMOR-1 LAAO with bulky side chains, Phe229, Leu361, and Gln535, are substituted by the less bulky Asn226, Gly342, and Thr450 residues in SvLAAO. This is consistent with *P. luteoviolacea* CPMOR-1 LAAO having its highest activity for Leu, Met, and Gln, which are relatively linear substrates that fit better in the site than the aromatic amino acids that are the preferred substrates for SvLAAO. The amino acid differences in the Lys-specific TvLAAO that endow it with specificity for Lys are more evident (Figure 8C). The substrate-binding site contains two residues, Asp212 and Asp315, that correspond to Leu245 and Gln359 in PlLAAO-1. These negatively charged side chains can interact with the positively charged side chain of the Lys substrate.

### 2.4. Phylogenetic Analysis of LAAOs

Using as a query the sequence of *P. luteoviolacea* CPMOR-1 LAAO in a BLAST search, the 60 most similar proteins were collected and subjected to phylogenetic analysis. Included in this analysis were representative LAAOs from the different types of species that have been previously described [18]. Two have been discussed here. The SvLAAO from *C. rhodostoma* venom is an example from a vertebrate. The TvLAAO from *T. viride* is from a fungus. Additionally included is the LAAO from *A. californica*, which is a gastropod. It was observed that these enzymes from non-bacterial sources were distantly related to the proteins most similar to *P. luteoviolacea* CPMOR-1 LAAO (Figure 9). All of the proteins detected in the BLAST are synthesized by marine gammaproteobacteria, except for one synthesized by the bacteriodete *Roseivirga pacifica*. Most of the bacteria synthesizing these proteins belong to the genus *Pseudoalteromonas*, and they are distributed in different branches. One of those branches contains the proteins from *P. phenolica* and other species such as *P. citrea*. A second branch contains many proteins from *P. piscicida*. There is a branch that splits into two groups: one containing proteins from *P. rubra* and the other containing proteins from *P. luteoviolacea*. The latter branch could be further divided. In fact, the proteins from *P. luteoviolacea* CPMOR-1 and CPMOR-2 reside in two different branches that are clearly separated according to the phylogenetic analysis performed in this study. These observations suggest that, even within the *Pseudoalteromonas* genus, there can be proteins with different enzymatic properties.

## 3. Discussion

### 3.1. Biological Roles of LAAOs

LAAOs are found in prokaryotes and eukaryotes, including animals. However, they are not typically metabolic enzymes. A role in nitrogen metabolism has only been proposed for some fungal LAAOs [33,34]. The metabolism of amino acids is usually initiated by pyridoxal phosphate-dependent transaminases that reversibly convert the amino acid substrate to the corresponding keto acid without additional products [35]. LAAOs also generate the corresponding keto acid, but release of the ammonium group is accompanied by the production of potentially deleterious H_2_O_2_ as well. The release of H_2_O_2_ has facilitated the adaptation of LAAO for other purposes that involve reaction with this metabolite. It is a component of snake venom, which uses it to induce apoptosis for toxicity. It also has generated interest in possible use as cancer treatments [36]. Some animals use it to inhibit bacterial growth [37]. In bacteria, LAAOs have been adapted for other purposes that involve reaction with H_2_O_2_. The antibiotic properties of H_2_O_2_ have been exploited by bacteria that are present in biofilms. Sessile surface bacteria are found in biofilms as both mono-species or, more often, multi-species communities, where they chemically cooperate, communicate, and compete [38,39]. The release of LAAOs into the biofilm allows them to use amino acids that are present to generate H_2_O_2_. LAAOs can provide a means for culling the population to increase the biofilm viability with limited resources, and the dispersal of biofilm sections into planktonic phase bacteria that can travel to colonize new surfaces [12]. *P. luteoviolacea* is a biofilm-inhabiting marine bacterium that produces an LAAO with a broad substrate specificity that would be ideally suited for this biological function.

### 3.2. Structure–Function Correlations

The portion of the active site of *P. luteoviolacea* CPMOR-1 LAAO that interacts with the α carbon and its carboxyl and amino constituents is highly conserved across different species, which suggests that this configuration near the FAD cofactor is necessary for the α amino bond-oxidation chemistry. In each structure, an Arg guanidino and Tyr hydroxyl interact with the carboxyl group of the substrate amino acid. A conserved Lys is also present in each structure. It has been proposed that this Lys stabilizes a water for use in catalysis [31].

The major differences between these LAAOs are seen in the active site residues forming a cage around the side chain of their respective amino acid substrates. The SvLAAO prefers aromatic amino acid substrates. While *P. luteoviolacea* CPMOR-1 LAAO also reacts with Phe, Trp, and Tyr, it has greater activity with Leu, Met, and Gln, which are more linear hydrophobic residues. Although this suggests a differential effect on the binding of the amino acid substrates in the two LAAOs, the *K*_m_ values for these six amino acid substrates for *P. luteoviolacea* CPMOR-1 LAAO are similar (see Table 1). However, the *k*_cat_ values for the aromatic amino acids are approximately 10-fold less than Leu, Met, and Gln. A possible explanation for this is that the binding affinity of these six amino acid substrates is similar, but the amino acid residues in the active site interact with the aromatic side chain of the substrate such that it affects the position of the reactive α-carbon and its substituents relative to the FAD. The mechanisms for the reaction of the substrate with the FAD could potentially occur via a nucleophilic attack or hydride transfer mechanism. In either case, if the distance over which such reactions have to occur is a bit longer, then this could slow the rate of reaction and decrease *k*_cat_. A more straightforward structure–function correlation is the observation that TvLAAO, which is specific for Lys, has two Asp residues that correspond to Leu and Gln in *P. luteoviolacea* CPMOR-1 LAAO. These Asp residues can endow the specificity towards the positively charged lysine side chain. A similar substitution of Asp in the active site is seen in the LAAO from *A. californica* that is specific for Lys and Arg [9,10].

### 3.3. Potential Applications of the P. luteoviolacea CPMOR-1 LAAO

The results of this study provide a logical basis for efforts to engineer *P. luteoviolacea* CPMOR-1 LAAO to select for substrate amino acid substrates based on the compatibility of the size and shape of their side chain with the binding pocket residues. Several areas of biotechnology could benefit from engineered LAAOs [1]. There is a particular interest in broad-spectrum LAAOs, which could be used to remove L-amino acids from racemic mixtures of L- and D-amino acids that are generated during synthetic processes. To this end, an ancestral LAAO was designed, and showed selectivity for 13 L-amino acids [40]. *P. luteoviolacea* CPMOR-1 LAAO has naturally evolved to exhibit a very broad substrate specificity. The development of a single enzyme with robust activity for all L-amino acids is likely not possible, since the means of accommodating one class of amino acid substrates necessarily excludes or greatly impairs their ability to react with a different class. For biotechnology applications, robust LAAO activity towards all amino acids might be achieved using a cocktail of multiple LAAOs with different specificities that react well under similar reaction conditions. The use of *P. luteoviolacea* CPMOR-1 LAAO as a template could allow for a site-directed mutagenesis strategy to generate a relatively small family of LAAOs that could cover all L-amino acids. This enzyme has robust activity, and variants could be designed that would be specific for positively or negatively charged amino acids and react well under similar reaction conditions. Thus, native *P. luteoviolacea* CPMOR-1 LAAO plus a few designed variants could accomplish this task.

## 4. Materials and Methods

### 4.1. Expression and Purification of P. luteoviolacea CPMOR-1 LAAO

The gene that encodes this LAAO was amplified from genomic DNA from the *P*. *luteoviolacea* CPMOR-1 strain [25]. The primers used were AminoORCPMOR1Nde 5′-GGAAAAcaTATGACACATTACACATTTG-3′ and AminoORCPMOR1Bam 5′-CAAAAAGCGGgGATcCAAATTACATAG-3′. The mismatches appear in lower case and the restriction sites are underlined. The KOD DNA polymerase (Merck, Darmstadt, Germany) was used in the amplification. The PCR fragment was digested with *Nde*I and *Bam*HI and it was cloned into a Novagen pET15b plasmid digested with the same enzymes using the T4 DNA ligase from Invitrogen. In this construction, the recombinant gene is coupled to an N-terminal hexahistidine tag sequence. The plasmid was purified with a Zyppy Plasmid Miniprep Kit (Zymo Research, Irvine, CA, USA), and inserted into BL21 competent cells (Thermofisher, Waltham, MA, USA) for expression. BL21 cells containing the gene were grown in LB medium with ampicillin as the selection antibiotic to 0.6–0.8 optical density, then induced with 1 mM isopropyl β-D-1-thiogalactopyranoside for 4 h at 30 °C, or overnight at 25 °C, prior to being harvested. The cells were disrupted by sonication and the soluble extract was applied to an Ni-NTA affinity column. The protein eluted over a range of 60–300 mM imidazole. Elution fractions were visibly yellow, indicating the presence of flavin, particularly when concentrated. Protein size and purity were assessed by sodium dodecyl sulfate-polyacrylamide gel electrophoresis (SDS-PAGE) stained with Coomassie blue.

### 4.2. Steady-State Kinetics

The steady-state kinetic parameters for the amino acid oxidase activity were determined as reported previously for analysis of another LAAO, LodA [41]. The reaction mixture contained the amino acid substrate and from 1 to 0.01 μM LAAO. Formation of the H_2_O_2_ product was coupled to the reaction of 0.05 mM Amplex Red with 0.1 U/mL of horseradish peroxidase. Reactions were performed in 50 mM KPi, pH 7 at 25 °C under room air. Initial rates were determined from the increase in absorbance at 571 nm associated with the production of resorufin from Amplex Red using ε_571_ = 54,000 M^−1^·cm^−1^. Alternatively, for substrates that interfered with the Amplex Red coupled reaction, another coupled assay was used that was previously reported for the study of another LAAO, GoxA [24]. The reaction mixture for this assay, which is coupled to ammonia production, contained 20 μ/mL glutamate dehydrogenase, 5 mM α-ketoglutarate, and 250 μM NADH, in the same buffer and temperature conditions. Initial rates were determined from the decrease in NADH absorbance at 340 nm using ε_340_ = 6220 M^−1^·cm^−1^. This assay was used for Gly and Cys. Initial rates were normalized by subtracting the background activity of the reaction mixture in the absence of enzyme. This varied with the substrate. These initial rates were then plotted against the substrate concentrations and analyzed using the Michaelis–Menten equation (Equation (1)), where *v* is the initial rate, S is the substrate, and E is *P. luteoviolacea* CPMOR-1 LAAO:*v*/[E] = *k*_cat_ [S]/(*K*_m_ + [S])(1)

### 4.3. Phylogenetic Analysis of LAAOs

Genes encoding similar LAAOs were detected by BLAST using as a query the sequence from the CPMOR-1 strain (Accession number WP 063369592). The BLAST was performed at the NCBI server (Accession, 18 November 2021). The 60 most similar proteins were selected and the LAAOs synthesized by the gastropod *Aplysia californica* (accession number NP_001191570) and the fungus *Trichoderma viride* (accession number BAR88116), and the protein from the *Calloselasma rhodostoma* venom (accession number NP_CAB71136) were added as outgroups. Evolutionary analyses were conducted in MEGA X [33]. All the sequences were aligned by using the program MUSCLE integrated in MEGA X. Two kinds of phylogenetic analyses were performed. First, a phylogenetic tree was constructed by the neighbor-joining method. The evolutionary distances were computed using the p-distance method and are in the units of the number of AA differences per site. All ambiguous positions were removed for each sequence pair. The reliability of each node in the tree was estimated by bootstrap analysis with 500 replicates. The final display of the tree was obtained using iTOL [42]. A tree was also constructed by the maximum likelihood method based on the Poisson correction model. Initial tree(s) for the heuristic search were obtained automatically by applying the Neighbor-Join and BioNJ algorithms to a matrix of pairwise distances estimated using a JTT model, and then selecting the topology with the superior log likelihood value. All positions with less than 95% site coverage were eliminated. Specifically, fewer than 5% alignment gaps, missing data, and ambiguous bases were allowed at any position. The reliability of each node in the ML tree was estimated by bootstrap analysis with 50 replicates.

### 4.4. Structure Modeling and Sequence Alignment

Comparison of the amino acid sequences of the LAAOs from *P. luteoviolacea* CPMOR-1 and CPMOR-2 indicated that they exhibit an 86.8% identity and 93.7% similarity. The crystal structure of the latter was recently reported [27]. Given the high sequence similarity, it was possible to construct a homology model of the *P. luteoviolacea* CPMOR-1 LAAO. The structural homology model was made in the SWISS-MODEL workspace (swissmodel.expasy.org (accessed on 5 January 2022)) using the crystal structure of LAAO from strain CPMOR-2 (PDB 7OG2) as a template with 95% coverage. The model was built based on the target–template alignment using ProMod3 [43]. The Global Model Quality Estimate [44] of the model was 0.9 and the QMEAN Z-score was −0.35, indicating a high level of accuracy of the tertiary structure of the model. The FAD molecule from the template was kept in the final model.

The amino acid sequences of the proteins were aligned using the Constraint-based Multiple Alignment Tool (COBALT) [45] from the NCBI platform, Blastp workspace (blast.ncbi.nlm.nih.gov, accessed on 12 June 2022), keeping the default server configuration. Manual adjustments were made to match the alignment of the three-dimensional structures, which were aligned using PyMol [46].

## Figures and Tables

**Figure 1 molecules-27-04726-f001:**
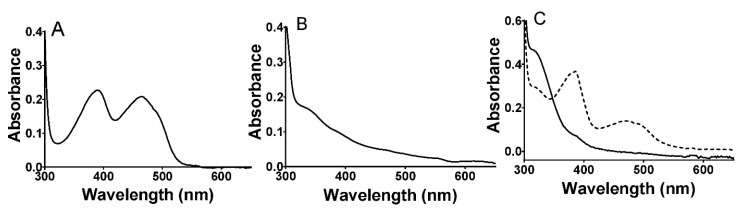
Absorbance spectra of *P. luteoviolacea* CPMOR-1 LAAO. (**A**). Spectrum of the as-isolated protein in the oxidized state. (**B**). Spectrum immediately after the addition of 1 mM methionine. (**C**). Spectra after the addition of a 20-fold excess of sodium dithionite. The dashed line spectrum was recorded immediately after the addition and the solid line spectrum was recorded approximately 8 min after mixing. Spectra were recorded of 30 μM LAAO in 50 mM potassium phosphate, pH 7.0.

**Figure 2 molecules-27-04726-f002:**
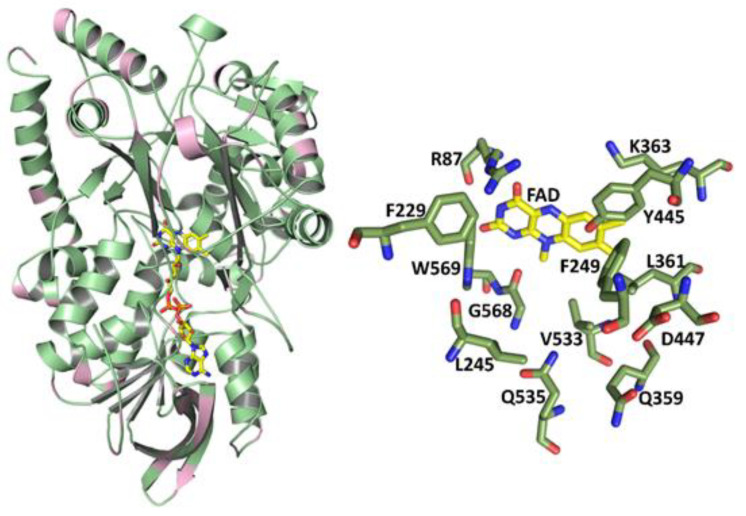
Structure model of *P. luteoviolacea* CPMOR-1 LAAO. The overall structure is shown as a cartoon on the **left**. The pink color indicates the regions in which differences in the amino acids between this enzyme and *P. luteoviolacea* CPMOR-2 LAAO were primarily located. FAD is shown with the carbons of the isoalloxazine and adenine rings yellow, nitrogens blue, oxygens red, and phosphorous orange. On the **right** are shown the residues in the substrate-binding area of the protein, which bind the amino acid substrates and position them for reaction with the FAD cofactor. Carbons of the amino acids are green.

**Figure 3 molecules-27-04726-f003:**
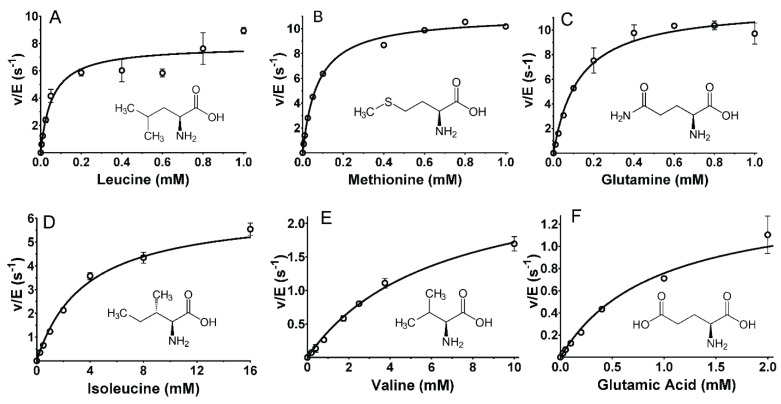
Steady-state kinetic analysis of the reactions of select amino acid substrates with *P. luteoviolacea* CPMOR-1 LAAO. The analysis of the data is shown for (**A**) Leucine, (**B**) Methionine, (**C**) Glutamine, (**D**) Isoleucine, (**E**) Valine and (**F**) Glutamic acid. The lines are the fits of the data to Equation (1). Data points are the average of three replicates. The structures of each amino acid are shown with each plot.

**Figure 4 molecules-27-04726-f004:**
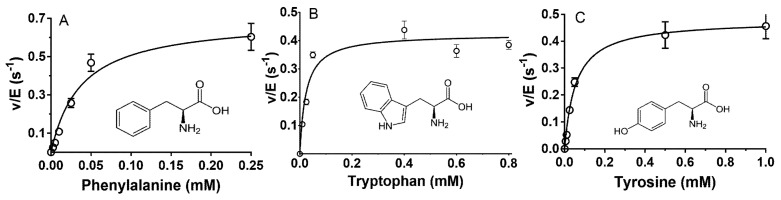
Steady-state kinetic analysis of the reactions of aromatic amino acid substrates with *P. luteoviolacea* CPMOR-1 LAAO. The analysis of the data is shown for (**A**) Phenylalanine, (**B**) Tryptophan and (**C**) Tyrosine. The lines are the fits of the data to Equation (1). Data points are the average of three replicates. The structures of each amino acid are shown with each plot.

**Figure 5 molecules-27-04726-f005:**
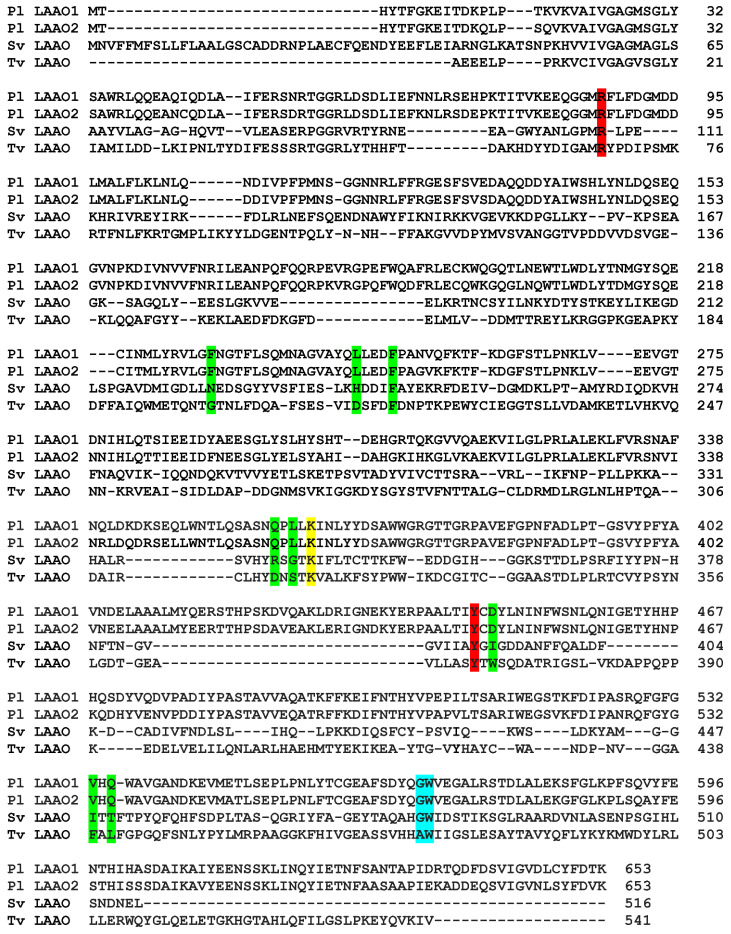
Amino acid sequence alignment of LAAOs. The sequences displayed are *P. luteoviolacea* CPMOR-1 LAAO (Pl LAAO1) (GenBank: WP_063369592), *P. luteoviolacea* CPMOR-2 LAAO (Pl LAAO2) (GenBank: KZN49687), SvLAAO (GenBank: CAB71136.), and TvLAAO (UniProt A0A0J9X1X3). Residues in the active site are highlighted. These are residues that interact with the carboxyl group (red) and amino group (cyan) bound to the α-carbon, residues that potentially interact with the amino acid side chain (green), and a residue that stabilizes a water molecule for use in catalysis (yellow). The numbering in this table corresponds to the amino acid numbers in the figures in this paper. It should be noted that the first 18 residues in the SvLAAO sequence comprise a signal sequence that is cleaved during export of the protein outside of the cell. The sequence in PDB 2IID does not include these residues. Therefore, there is an 18-residue difference in the numbering adopted here compared to the numbering in the PDB sequence.

**Figure 6 molecules-27-04726-f006:**
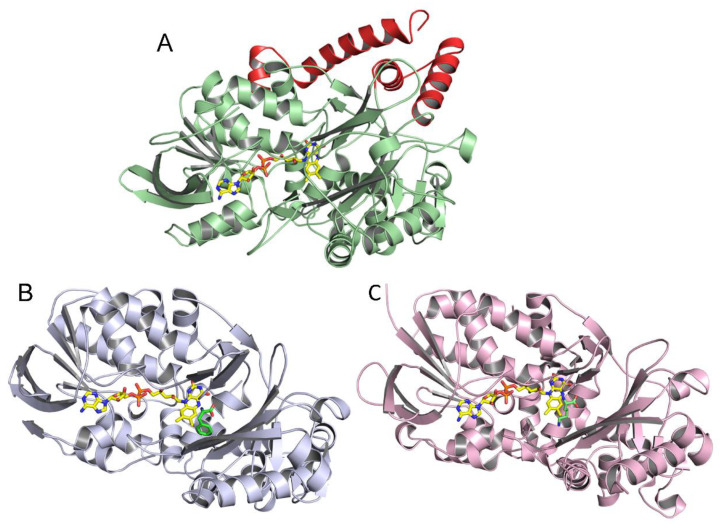
Monomeric structures of LAAOs. (**A**) *P. luteoviolacea* CPMOR-1 LAAO. (**B**) SvLAAO from *C. rhodostoma* with Phe substrate bound (PDB 2IID) [28]. (**C**) TvLAAO (PBD 7C3H) [29] with Lys substrate bound. FAD is shown with the carbons of the isoalloxazine and adenine rings yellow, nitrogens blue, oxygens red, and phosphorous orange. The carbons of the Phe and Lys substrates are colored green. The additional helices present in *P. luteoviolacea* CPMOR-1 LAAO (**A**) that are not in the other LAAOs are colored red.

**Figure 7 molecules-27-04726-f007:**
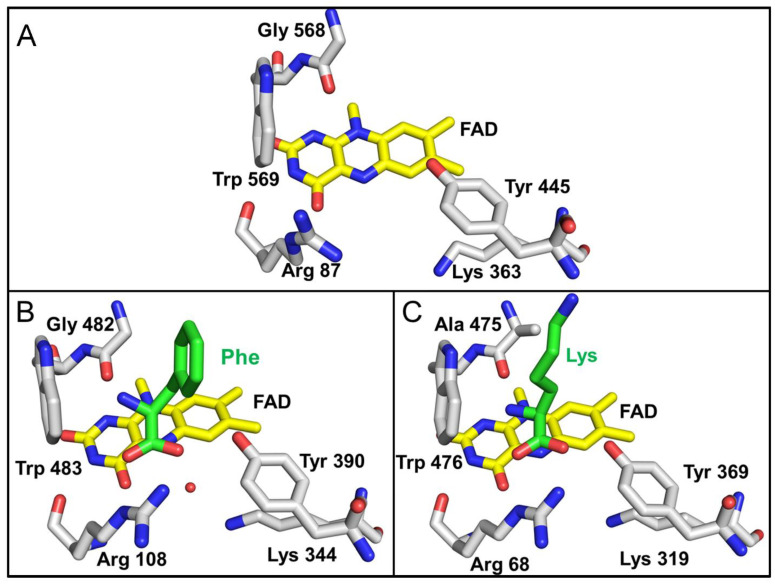
Active site residues that interact with the carboxyl and amino groups of the α-carbon of the amino acid substrate. (**A**) *P. luteoviolacea* CPMOR-1 LAAO. (**B**) SvLAAO from *C. rhodostoma* (PDB 2IID) with a Phe substrate present. The red sphere is a water molecule, which is coordinated by Lys344. (**C**) TvLAAO (PBD 7C3H) with a Lys substrate present. Oxygens are red, nitrogens are blue, and carbons are grey in the protein amino acids, yellow in FAD and green in the amino acid substrates.

**Figure 8 molecules-27-04726-f008:**
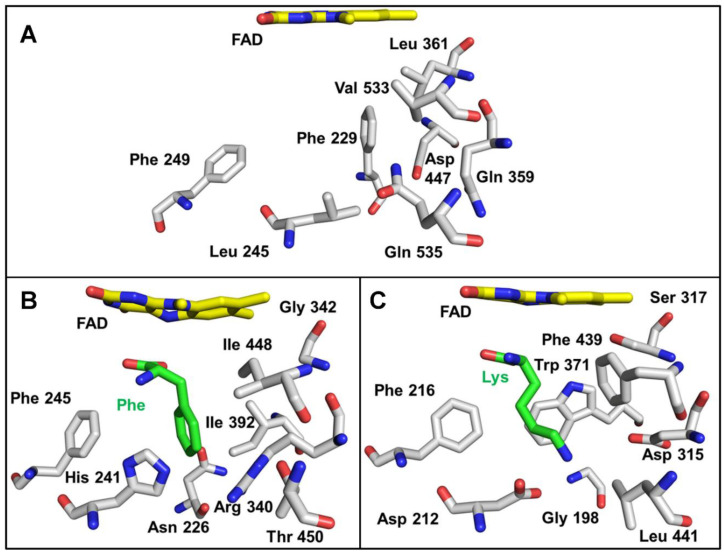
Active site residues that interact with the side chains of the amino acid substrate. In contrast to Figure 7, the residues in the active site pocket that potentially interact with the substrate side chain are shown. (**A**) *P. luteoviolacea* CPMOR-1 LAAO. (**B**) SvLAAO from *C. rhodostoma* (PDB 2IID) with a Phe substrate present. (**C**) TvLAAO (PBD 7C3H) with a Lys substrate present. Oxygens are red, nitrogens are blue, and carbons are grey in the protein amino acids, yellow in FAD and green in the amino acid substrates.

**Figure 9 molecules-27-04726-f009:**
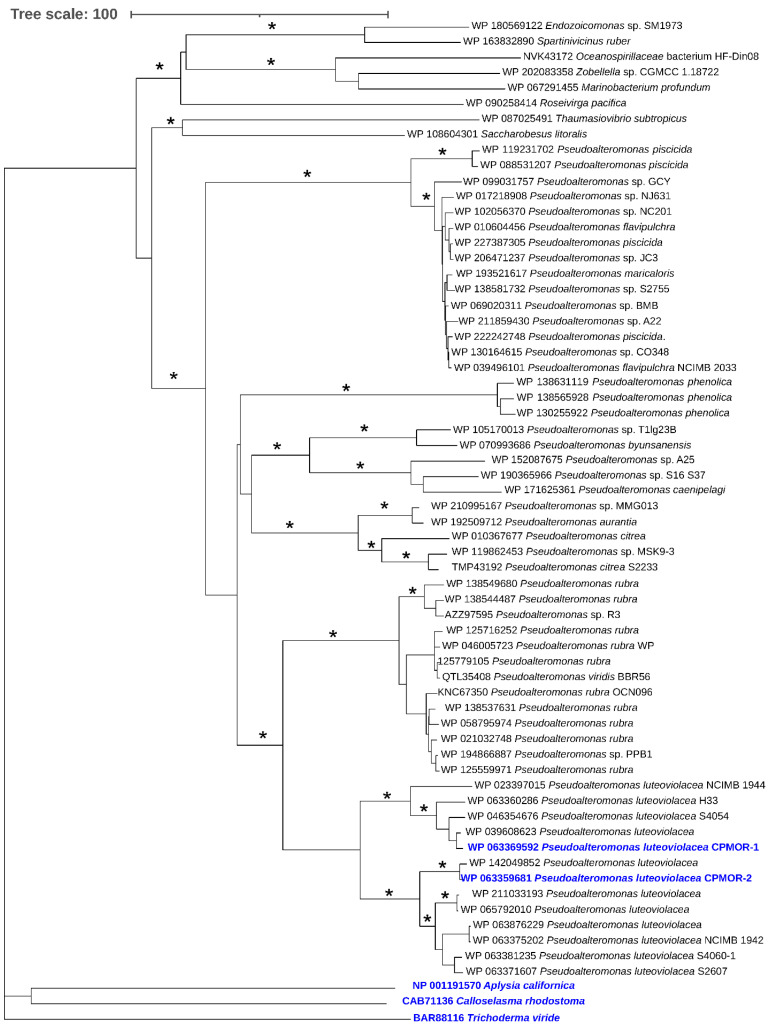
Phylogenetic relationship of proteins with similarity to PlLAAO. The tree shown was constructed using the neighbor-joining method integrated in the program MEGA X [32]. The tree is drawn to scale, with branch lengths in the same units as those of the evolutionary distances used to infer the phylogenetic tree. Asterisk at the branches indicate bootstrap values higher than 90% for the phylogenetic analyses performed with both the neighbor-joining and maximum likelihood methods. Colored blue are the *P. luteoviolacea* strains CPMOR-1 and CPMOR-2 and the proteins from the non-bacterial species.

**Table 1 molecules-27-04726-t001:** Steady-state kinetic parameters for the reaction of *P. luteoviolacea* CPMOR-1 LAAO with L-amino acid substrates. Each amino acid was assayed as described in the Section 4 at pH 7, 25 °C, and fit with Equation (1). Values are the average of at least three replicates. The R^2^ values for the fits of the data for each amino acid substrate are given.

Substrate	*k*_cat_ (s^−1^)	*K*_m_ (mM)	*k*_cat_/*K*_m_ (M^−1^·s^−1^)	R^2^
Leu	9.38 ± 0.21	0.0567 ± 0.0051	1.65 × 10^5^	0.995
Met	10.9 ± 0.4	0.0737 ± 0.0011	1.48 × 10^5^	0.99
Gln	12.0 ± 0.7	0.126 ± 0.025	9.51 × 10^4^	0.979
Phe	0.698 ± 0.052	0.0376 ± 0.0096	1.85 × 10^4^	0.969
Trp	0.50 ± 0.023	0.0565 ± 0.0123	8.85 × 10^3^	0.984
Tyr	0.478 ± 0.025	0.0552 ± 0.0109	8.67 × 10^3^	0.972
Ile	6.39 ± 0.34	3.65 ± 0.61	1.75 × 10^3^	0.986
Glu	1.45 ± 0.12	0.921 ± 0.235	1.58 × 10^3^	0.967
Arg	1.07 ± 0.05	2.11 ± 0.38	5.08 × 10^2^	0.967
Val	2.80 ± 0.28	6.28 ± 1.17	4.45 × 10^2^	0.981
Ala	5.26 ± 1.13	21.7 ± 0.83	2.42 × 10^2^	0.99
Cys	0.113 ± 0.012	0.467 ± 0.215	2.41 × 10^2^	0.965
Lys	5.10 ± 0.27	43.1 ± 7.6	1.18 × 10^2^	0.969
His	0.108 ± 0.008	1.01 ± 0.30	1.07 × 10^2^	0.978
Asn	0.259 ± 0.022	7.96 ± 1.63	32.5	0.969
Ser	0.281 ± 0.032	40.3 ± 9.81	6.97	0.986
Thr	0.135 ± 0.008	173 ± 29	0.777	0.984
Asp	1.72 ± 0.98 × 10^−3^	14.3 ± 27.6	0.12	0.832
Gly	5.45 ± 0.55 × 10^−3^	71.1 ± 31.4	7.67 × 10^−2^	0.923

**Table 2 molecules-27-04726-t002:** Steady-state kinetic parameters for the reaction of *P. luteoviolacea* CPMOR-1 LAAO with L-amino acid substrates. Each amino acid was assayed as described in the Section 4 at pH 7, 25 °C, and fit with Equation (1). Values are the average of at least three replicates. The R^2^ values for the fits of the data for each amino acid substrate are given.

Substrate	*k*_cat_ (s^−1^)	*K*_m_ (mM)	*k*_cat_/*K*_m_ (M^−1^·s^−1^)	R^2^
Trp	0.50 ± 0.023	0.0565 ± 0.0123	8.85 × 10^3^	0.984
5-OH Trp	0.371 ± 0.056	0.462 ± 1.97	8.04 × 10^2^	0.935
Tyr	0.478 ± 0.025	0.0552 ± 0.0109	8.67 × 10^3^	0.972
3-Nitro Tyr	0.545 ± 0.045	0.038.6 ± 10	1.41 × 10^4^	0.981
Cys	0.113 ± 0.012	0.467 ± 0.215	2.41 × 10^2^	0.965
Homo Cys	0.179 ± 0.062	0.306 ± 0.238	5.84 × 10^2^	0.982

## Data Availability

The authors declare that the main data supporting the findings of this study are available within this article.

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
