# Peer review of "Structural Determinants of the Specific Activities of an L-Amino Acid Oxidase from Pseudoalteromonas luteoviolacea CPMOR-1 with Broad Substrate Specificity"

_molecules, 2022, doi:10.3390/molecules27154726_

Round 1

Reviewer 1 Report

This is a good piece of work. The authors determined the kinetic parameters of recombinant LAAO of P. luteoviolacea CPMOR-1, described the substrate preference of this enzyme, and compared its active site residues with those of other enzymes. The experimental methods and results are robust and support the conclusions. However, there are several areas for improvement.

(1) The results of kinetic parameters and crystal structure of the CPMOR-2 LAAO have already been reported. No comparison with those results has been made. Therefore, it is not clear to the readers what is a new result or new finding. Indeed, the amino acid sequences are not 100% identical. However, they are very similar (86.8% identity and 93.7% similarity). So what makes them different in terms of their properties?

(2) The authors have compared the three-dimensional structures of other species to clarify substrate preferences, but ultimately it is unclear which amino acid residues in CPMOR-1 LAAO are important in determining the substrate preference of the enzyme. How about comparing CPMOR-1 LAAO to CPMOR-2 LAAO, which has a very similar amino acid sequence? Also, why not create mutants of amino acid residues that the authors consider important and examine their activity?

Others

(1) Line 30, “coSancheznserved”: “conserved”

(2) Lines 32-33, “For P. luteoviolacea CPMOR-1…such as deracemization”: The meaning of this statement is unclear.

(3) Line 97, “The purified LAAO ran as a single band on SDS-PAGE”: Since there is no figure, the reader cannot determine whether it is a single band or not. It is good to attach the figure of the electrophoresis.

(4) Lines 97-98, “molecular weight”: “molecular mass”

(5) Figures 2 and 3: The structure of amino acids is common knowledge to the readers. The parameters obtained from the curves are in Table 1. Thus, figures 2 and 3 are not needed at all.

(6) Lines 144-146, “Leu has a 3-carbon…with an amide (Figure 2C)”: The description of the characteristics of the side chains of Leu, Met, and Gln is not necessary here. This is because the readers know the structures of these side chains and there are no results showing a strong correlation between those features and activity or their relationship to the active site structure.

(7) Figure 4: It is difficult to know the degree of similarity without adding an asterisk or other mark under the amino acid sequences.

(8) Lines 227-229: With what degree of reliability was this three-dimensional structure created?

(9) Lines 237-249: Where are the two major structural differences in this three-dimensional structure (Figure 5)?

(10) Line 238, “Lys410”: Leu410?

(11) Line 276, “bulky”: How about a comparison of the surface model or the size of the active site. It is difficult to understand the characteristics of the active sites from these stick models.

(12) Figures 4 and 5: A comparison of P. luteoviolacea CPMOR-1 and CPMOR-2 LAAOs would provide more useful information on the substrate specificity of the CPMOR-1 LAAO.

(13) Lines 307-309, “Significantly, the proteins…large evolutionally distance”: The amino acid sequence shows an 86.8% identities between the two P. luteoviolacea CPMOR LAAOs. In which areas are there major evolutionary differences? How similar are the active sites between the two enzymes? How similar are the substrate specificities and kinetic parameters between the two enzymes?

Reviewer 2 Report

The manuscript “Structural determinants of the specific activities of an L-aminoacid oxidase from Pseudoalteromonas luteoviolacea CPMOR-1 with broad substrate specificity” describes the catalytic activity of CPMOR-1 enzyme towards 20 proteinogenic aminoacids and 3 aminoacid derivatives. The authors present KM and kcat values, according to which a broad substrate specificity is evident.

The protein structural features which are related to the preference toward specific substrate is defined, and a phylogenetic analysis to detect a structurally related proteins is performed.

Overall the article is well-written and the kinetic parameters of the enzymatic activity are useful experimental data points, which can be used elsewhere, e.g. for the development of the models for enzymatic activity.

However, I feel the authors have not answered all the questions which arise from the experimental data they obtained. The most interesting point is a large difference between the enzymatic activity towards Leu and Ile, and also between Asn and Gln, while the activity towards Ser and Thr is close. Why the authors didn’t perform the analysis of the conformation of these aminoacids (and possibly Val) in the active site, as presented in Figures 6 and 7?

There are also small typos within the text:

Line 4: “1with” a space is needed

Line 30: must be “conserved”

Lines 276-278: the sentence is unclear

439: must be “coupled”

Overall I recommend the article for publication after these minor points are addressed.

Reviewer 3 Report

The manuscript reports a study on the L-amino acid oxidase (LAAO) of CPMOR-1. The steady-state kinetics of the expressed and purified protein was used to characterize its reactivity towards amino acids, and the differences in activity were correlated with the structural characteristics in the LAAO active sites. Structural comparison of LAAO of CPMOR-1 and CPMOR-2 was performed using empirical (X-ray) data for LAAO (SvLAAO and TvLAAO) from CPMOR-2 and  homology model of LAAO from CPMOR- 1. And here I have a fundamental remark:

Using a very approximate homology model for comparison with structural data is not the right approach. Each homology model must be carefully refined by molecular dynamics (MD) simulations to obtain the well-equilibrated conformation that should be used for comparison. The MD simulation running is quasi-automatic and can be easily realised by the non-expert in the field, but this step is absolutely necessary to obtain a usable model for comparison, even qualitative. Other remarks:

1. Results presented as graphs (Fig. 1 - 3) will be much better and clearer for comparison if presented on the same scale.

2. Figure 5: The difference between the proteins will be more apparent when the compared proteins are superimposed.

3. In the legends of figures 5 to 7, the various images A, B, C will be designated by (A), (B) and (C).

Round 2

Reviewer 3 Report

Read, please,  my provious remarks.

Author Response

The previous remarks were fully addressed in our previous response. The reliability of our model is documented in the revised paper and the few other points were addressed.